# RedSync : Reducing Synchronization Traffic for Distributed Deep Learning

## Abstract

Data parallelism has become a dominant method to scale Deep Neural Network (DNN) training across multiple nodes. Since the synchronization of the local models or gradients can be a bottleneck for large-scale distributed training, compressing communication traffic has gained widespread attention recently. Among several recent proposed compression algorithms, Residual Gradient Compression (RGC) is one of the most successful approaches—it can significantly compress the transmitting message size (0.1% of the gradient size) of each node and still preserve accuracy. However, the literature on compressing deep networks focuses almost exclusively on achieving good compression rate, while the efficiency of RGC in real implementation has been less investigated. In this paper, we develop an RGC method that achieves significant training time improvement in real-world multi-GPU systems. Our proposed RGC system design called RedSync, introduces a set of optimizations to reduce communication bandwidth while introducing limited overhead. We examine the performance of RedSync on two different multiple GPU platforms, including a supercomputer and a multi-card server. Our test cases include image classification on Cifar10 and ImageNet, and language modeling tasks on Penn Treebank and Wiki2 datasets. For DNNs featured with high communication to computation ratio, which has long been considered with poor scalability, RedSync shows significant performance improvement.

## 1 Introduction

For training large-scale deep neural networks (DNNs) on multiple computing nodes, data parallelism has emerged as the most popular choice due to its simplicity and effectiveness (Dean et al. (2012); Recht et al. (2011)). However, the communication bandwidth of network fabric has become the bottleneck limiting data parallel performance. On one hand, models of DNNs, which already contain tens to hundreds of layers and totaling 10-20 million parameters today, continue to grow bigger. Therefore, the requirement of communicating model parameter updates among all computing nodes poses a higher challenge to network bandwidth. On the other hand, the development of DNN training accelerators has shifted the bottleneck of training towards communication across models. As the evolution of the inter-connected network bandwidth is not as fast as computing hardware, synchronization overhead has become the bottleneck of data parallelism on distributed systems using new computing hardware.

Many recent studies focused on reducing the communication cost between nodes by reducing the size of the gradients to be transmitted. One line of work (Seide et al. (2014); Alistarh et al. (2017); Wen et al. (2017)) propose to quantize the gradients to low-precision values. Considering compression ratio (ratio of compressed gradients size to their original size) achieved by quantization is limited, another line of research orthogonal to quantization is to sparsify communication gradients and restrict weight-updates to a small subset of parameters. Residual Gradient Compression (RGC) method (Strom (2015); Aji & Heafield (2017); Chen et al. (2017); Lin et al. (2017); Sattler et al. (2018)) is currently the most promising pruning method to achieve good compression ratio while ensuring no loss of training accuracy. It transmits only a small subset of gradients and maintains the remaining gradients locally as residuals to be added to gradients of the next iteration. The first RGC implementation is proposed by Strom (2015) and uses a threshold-based method to only send gradients larger than a predefined constant threshold for fully-connected layers. Considering a predefined threshold is hard to be chosen appropriately, Aji & Heafield (2017) improve the robustness

of RGC by selecting top 1% gradients to communicate according to their magnitude. Because these two implementations are tuned for some specific network structures, applying them to other DNNs will lead to accuracy loss as indicated in Chen et al. (2017). Based on their work, the latest RGC variants, such as (Sattler et al. (2018); Chen et al. (2017); Lin et al. (2017)), are able to achieve a 0.1% compression ratio on local gradients while ensuring almost no loss of model accuracy on a variety of DNN structures after introducing some key modifications.

Despite of good model accuracy achieved with simulation experiments, no recent studies have discussed the potential performance gain after integrating the latest RCG methods to real distributed training system, especially to the multi-GPU systems equipped with high-quality network infrastructures. The challenges of applying RGC to distributed GPU systems come from two aspects. First, there is no efficient compression algorithm proposed for RGC method. According to our experimental results, selecting top-0.1% elements with the state-of-the-art GPU-based top-k algorithm are so expensive that the overhead of compression is much higher than the benefits of network bandwidth reduction. Second, synchronization of sparse data structures is nontrivial to be supported with existing efficient communication libraries, such as Message Passing Interface (MPI), which are designed for dense data structures.

Targeting multi-GPU systems, a highly-efficient RGC implementation called RedSync is proposed. Our contributions are listed as follows:

- We combined pruning and quantization techniques together to compress transmitting gradients. A set of parallel-friendly top-0.1% selection methods are designed to support pruning operations inside GPU device memory, which are orders of magnitude faster than the state-of-the-art GPU-based top-k selection method.
- Considering the distribution characteristics of communication data, we apply allgather operation using MPI for a sparse synchronization scheme. A cost model is derived to analyze both communication cost and calculation overhead. Based on it, we pointed out potential performance gain and the bottleneck of our implementation.
- RedSync is able to ensure almost no accuracy loss to train a set of DNNs after integrating with the latest algorithm improvements. This is the first work, as far as we known, to evaluate the performance of RGC method on the scale of 128 GPUs. RedSync provides significant performance improvements for communication-intensive networks, like VGG, AlexNet and some LSTMs.

## 2 DESIGN AND IMPLEMENTATION OF REDSYNC

We first give an overview of a simple RGC workflow used in RedSync (see more details in Algorithm 1). We denote a DNN model as $f(\mathbf{w})$, where $\mathbf{w}$ is the vector of parameters. We assume a system has $N$ workers. Each worker, say the $k$-th worker, holds a local dataset $\chi_k^t$ at iteration $t$ with size $b$ and a local copy of the global weight $\mathbf{w}$. Synchronous SGD method is adopted in RedSync. At each iteration, node $k$ computes the gradient $G^k$ using local data, where $G_j^k$ indicates gradients of layer $j$. Each node also maintains a residual $V^k$, which is initialized as $0$ and used to accumulate untransmitted gradient from previous iterations. After added with latest gradient, a subset of residuals is selected as the *communication-set*, and is compressed into sparse data structures. The `select` operation in Algorithm 1 chooses more important elements based on magnitude. Those selected elements (denoted ask Masks) are synchronized among all the nodes using allreduce operations, which is able to take advantage of the highly-optimized allreduce operation on HPC systems (Thakur et al. (2005)). Synchronous SGD

---

**Algorithm 1** Residual Gradient Compression

**Input:** node id $k$, the number of node $N$
**Input:** dataset $\chi$
**Input:** mini batch size $b$ per node
**Input:** initial model $w = w[0], ..., w[\#layer]$
**Input:** compression ratio $D$
  $V^k \leftarrow 0$
  **for** $t = 0, 1, ...max\_iter$ **do**
    sample $b$ elements as $\chi_k^t$
    $G^k \leftarrow \nabla f(\chi_k^t ; \mathbf{w})$ by forward and backward propagation
    **for** $j = \#layer, \#layer - 1, ..., 0$ **do**
      $V_j^k += G_j^k$
      Masks $\leftarrow$ `select` $(V_j^k, D)$
      $G_j^k \leftarrow$ `Allreduce(compress(`$V_j^k \cdot$ Masks`))`
      $V_j^k \leftarrow V_j^k \odot (1 - $ Masks$)$
    **end for**
    $\mathbf{w} \leftarrow$ SGD$(\mathbf{w},$ `decompress(`$G^k$`))`
  **end for**

---

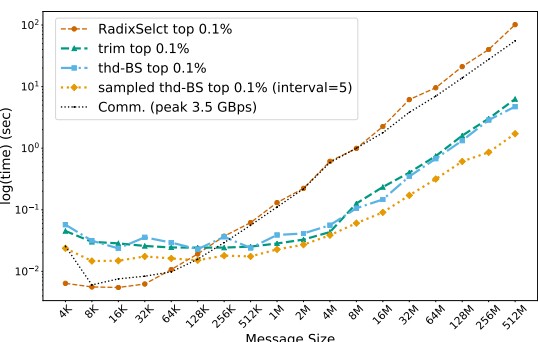

Figure 1: Performance of four communication-set selection methods under message sizes. Elements in the data list are generated randomly from a standard uniform distribution. `Comm.` illustrates the time taken to synchronize the message through a network with a peak bandwidth of 3.5GBps by allreduce operation. Performance is measured as total time cost for 100 times independent operations.

implemented with allreduce has been widely adopted in state-of-the-art large-scale CNN training tasks (Goyal et al. (2017) and You et al. (2017)). Remaining elements outside the communication-set are assigned as new residuals of the next iteration. The workflow of this algorithm is the same as an RGC variant called Deep Gradient Compression Method mentioned Lin et al. (2017). In the following, we details our contribution in implementations of `select`, `Allreduce` and `decompress` to make this workflow efficient in practice.

## 2.1 PARALLEL-FRIENDLY COMPRESSION

The efficiency of communication-set selection method is critical for the RGC system's overall performance. Since a predefined threshold is difficult to determine, recent work (Lin et al. (2017); Sattler et al. (2018)) suggest to select top $0.1\%$ elements from residuals of each layer as the communication-set. However, the top-$0.1\%$ selection is nontrivial to be implemented on GPU. One of the most efficient top-$k$ selection methods designed for GPU can be implemented based on *radixSelect* algorithm (Alabi et al. (2012)), which determines each bit of the $k$-th largest element by scan and scatter. Serial *scan* (Sengupta et al. (2007)) and scatter operations are extremely time-consuming. As shown in Figure 1, the computation time for top-0.1% with radixSelect on a Titan X GPU sometimes is even slightly higher than the time for synchronizing these parameters through a 3.5 GBps network. To avoid performing a top-0.1% operation on a large number of parameters, we propose two communication-set selection algorithms called *trimmed top-k selection* and *threshold binary search selection*, which are more efficient on GPUs.

**Trimmed top-$k$ selection.** Observing that the distribution of residuals is usually similar to a normal distribution, we can use statistical features to remove most of the smaller elements and limit radixSelect operation on a relatively small subset. As shown in Algorithm 2, we first calculate the mean and maximum of residuals' absolute values of this layer. A relative large threshold value is chosen according to mean and maximum value, for example, $0.8 \times (max - mean) + mean$. Operation `count_nonzero` gets the number of elements whose absolute values are greater than the threshold. If the number is smaller than $k$ (the number of top-0.1% elements ), we dynamically decrease the threshold until we find the number of parameters whose absolute value above the threshold is larger than $k$. Then we trim all elements that are less than the threshold and perform a top-$k$ selection operation using radixSelect on the remaining elements. Operation `mean`, `max` and `count_nonzero` can all be efficiently implemented with a single reduction operation. `nonzero_indices` is a typical *stream compaction* problem, which uses just one scan operation as its backbone (Sengupta et al. (2006)).

**Threshold binary search selection.** For some layers with very large numbers of parameter elements, even conducting radixSelect on a small subset of elements will still be a very time-consuming operation. In order to completely avoid using radixSelect operation on GPU, we propose a method to select approximate top-0.1% elements as communication-set. Instead of identifying the $k$th (top 0.1%th) largest element, we search for a threshold to make it between the $k$th to $2k$th largest element, and then select elements larger than the threshold as communication-set. In this case, at least 0.1% largest elements are included in the communication-set. As shown in Algorithm 3, we use a binary search algorithm to find such a threshold. To avoid excessive searching, it will always be terminated when the difference of left bound and right bound is less than a small value $\epsilon$.

**Algorithm 2** trimmed top-$k$ Selection

**Input:** tensor to be compressed $X$
**Input:** number of elements remained $k$
**Output:** $< indice, values >$
1: $mean \leftarrow \texttt{mean}(\text{abs}(X))$
2: $max \leftarrow \texttt{max}(\text{abs}(X))$
3: $\epsilon \leftarrow 0.2$
4: $ratio \leftarrow (1 - \epsilon)$
5: $nnz = \texttt{count\_nonzero}(\text{abs}(X) > threshold)$
6: **while** $nnz > k$ **do**
7: $\quad threshold \leftarrow mean + ratio \times (max - mean)$
8: $\quad nnz = \texttt{count\_nonzero}(\text{abs}(X) > threshold)$
9: $\quad ratio = ratio - \epsilon$
10: **end while**
11: $indice \leftarrow \texttt{nonzero\_indices}(\text{abs}(X) > threshold))$
12: $values \leftarrow X[indice]$

**Algorithm 3** Top-$k$ selection with threshold binary search selection

**Input:** tensor to be compressed $X$
**Input:** number of elements remained $k$
**Input:** Termination condition parameter $\epsilon$
**Output:** $< indice, values >$
1: $mean \leftarrow \texttt{mean}(\text{abs}(X)); max \leftarrow \texttt{max}(\text{abs}(X))$
2: $l \leftarrow 0.0; r \leftarrow 1.0; threshold = 0.0$
3: **while** $r - l > \epsilon$ **do**
4: $\quad ratio = l + (r - l)/2$
5: $\quad threshold \leftarrow mean + ratio \times (max - mean)$
6: $\quad nnz = \texttt{count\_nonzero}(\text{abs}(X) > threshold)$
7: $\quad$ **if** $nnz > k$ and $2k > nnz$ **then**
8: $\quad\quad$ break
9: $\quad$ **else if** $nnz < k/2$ **then**
10: $\quad\quad r = threshold$
11: $\quad$ **else**
12: $\quad\quad l = threshold$
13: $\quad$ **end if**
14: **end while**
15: $indice \leftarrow \texttt{nonzero\_indices}(\text{abs}(X) > threshold))$
16: $values \leftarrow X[indice]$

For layers with large sizes, such as the first fully-connected layer in VGG16 and softmax layer in LSTM, the time for `count_nonzero` operation is still not negligible. We further improve the efficiency of the selection algorithm by reducing the number of `count_nonzero` operations. We recommend that, after a threshold binary search for this layer, the threshold element can be reused in the next few iterations. The interval of search is empirically set to 5, and the selection algorithm introduces only one `nonzero_count` overhead on average.

In Figure 1, we compared the time cost of different selection approaches on parameter lists of different sizes. Compared with directly performing radixSelect, both proposed methods significantly reduce the selection time for large sizes. For top-0.1% selection on 64MB elements, trimmed top-k and sampled threshold binary search selection are 38.13 and 16.17 × faster than radixSelect. In practice, we dynamically choose compression strategies: For smaller parameter sets such as biases and batch norm layers, we do not compress residuals or directly use radixSelect to select top-0.1% significant elements. Trimmed top-k selection is suitable for parameters of middle size layers, like convolutional layers, because it can ensure the compression ratio to be exactly 0.1% and introduce no extra communication bandwidth requirements. Threshold binary search based selection is suitable for large size layers, like hidden layers and softmax layers in LSTMs, for which the compression cost is more critical to be optimized than the communication cost.

### 2.1.1 QUANTIZATION OF COMPRESSED RESIDUALS

Compressed residuals should include $k$ indices and $k$ values. We further investigate the possibility of quantizing these values. By setting the values of all elements of the same sign in the communication-set to their mean, we can almost eliminate the communication bandwidth requirement of value information transmitting by using only one floating-point number instead of $k$. In order to facilitate quantization compression, we slightly modify our `select` method to ensure that elements in the communication-set are all of the same sign. It can be achieved by choosing the largest $k$ elements and the smallest $k$ elements as communication-set in turns. In other words, if we select the largest $k$ elements (all positive numbers) in this layer as the communication-set at current iteration, we will choose smallest $k$ elements (all negative numbers) as the communication-set for the next iteration. It is worth noting that *sampled threshold binary search selection* cannot be used with quantization. In addition, we do not quantify the output layer of the DNN, in order to distinguish the correct classification information.

### 2.2 SPARSE SYNCHRONIZATION AND DECOMPRESSION

Synchronization of dense gradient structures in traditional distributed DNN systems can be simply implemented with an allreduce operation, which has been well-studied on multiple-GPU systems (Awan et al. (2017)). However, the design of a sparse allreduce in a distributed setting is not as simple because each worker may contribute different non-zero indices in its compressed residuals.

According to our observation, there are very few overlapping indices of the communication-set distribution of different nodes. For example, training VGG16 on Cifar10 dataset using 16 GPUs with a compression ratio as 0.1% for each node, the averaged compression ratio of synchronized residuals of all nodes is 1.55%. We utilize the allgather operation, an operation in which the data contributed by each node is gathered at all nodes, to implement sparse allreduce. The message representing compressed residuals of each node should include the information of indices and values of elements in communication-set. When using threshold binary search selection, the length of each node's message is different. As a result, the packaged message should also include an initial element, which indicates the length of the compressed elements. Instead of using two allgather operations for indices and values message separately, we package the indices and values into a single message to reduce latency.

After finishing the allgather operation, each node collects $N$ compressed residuals of this layer from all the other nodes. We add the compressed residuals to the corresponding weights in the local model after scaling with the learning rate. It can be seen as an operation that adds a sparse array to a dense array, which has been fully-optimized in Level 1 function axpyi() of cuSparse library on GPU.

## 2.3 OTHER TECHNIQUES

RedSync implements a set of algorithm improvement techniques proposed in Lin et al. (2017). We details momentum correction, momentum factor masking and our modification to warmup training in Appendix C, as well as local gradient clipping in Appendix B.

## 2.4 PERFORMANCE MODEL FOR RGC COMMUNICATION

To analyze the potential performance gain of sparse synchronization, we adopt a widely-used performance model to estimate the communication cost in terms of latency and bandwidth used. We assume that the time taken to send a message between any two nodes can be modeled as $\alpha + n\beta$, where $\alpha$ is the latency (or startup time) per message, independent of message size, $\beta$ is the transfer time per byte, and $n$ is the number of bytes transferred. The node's network interface is assumed to be single ported; i.e. at most one message can be sent and one message can be received simultaneously. $M$ is the number of elements in residuals of current layer. $D$ is the compression ratio. In the case of reduction operations, we assume that $\gamma_2$ is the computational cost for performing the reduction operation for a message of size $M$, and $\gamma_1$ is the cost to decompress the collected sparse message of size $M$. For the case where the compression ratio of each node is different, which is always true for the binary search method, $D$ represents the average compression ratio of all nodes.

Suppose that we use recursive doubling for allgather and Rabenseifners algorithm mentioned in Thakur et al. (2005) for allreduce communication. The cost of quantized sparse and dense synchronization is illustrated Equation 1 and 2, respectively. The derivations are left in Appendix A.

$$T_{sparse} = T_{select} + \log(p)\alpha + (p-1)(MD)\beta + p\gamma_1 \quad (1) \qquad T_{dense} = 2\log(p)\alpha + 2\frac{p-1}{p}M\beta + \frac{p-1}{p}\gamma_2 \quad (2)$$

As implicated by the performance model, **the compression rate for the model is not equal to the compression rate for communication bandwidth.** The bandwidth term of sparse synchronization is $(p-1)DM\beta$, which is proportional to the number of nodes $p$. Even if the sparseness $D$ is 0.1% for all $p$ node, when $p$ is 128, the communication bandwidth for sparse synchronization will be 12.8% of dense synchronization rather than 0.1% of dense synchronization. Second, **the overhead of reduction may be a new bottleneck when scaling RedSync to larger scale.** The last term $p\gamma_1$ in Eq. 1 indicates that the overhead to do reduction also increases linearly with the number of nodes $p$. However, in Eq. 2, reduction overhead almost does not increase with number of nodes.

## 3 EXPERIMENTAL RESULTS

### 3.1 SETUPS

We tested the accuracy and performance of our proposed implementation on two different multi-GPU systems, including a world's top GPU supercomputer and a multi-GPU server. **Muradin** is a

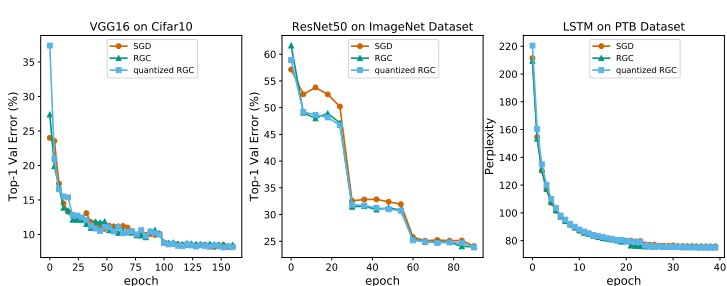

Figure 2: Left : top-1 validation accuracy vs number of epochs of training VGG16 on Cifar10 (4 GPUs, total batch size = 256). Center : top-1 validation accuracy vs number of epochs of training ResNet50 on ImageNet (8 GPUs, total batch size = 256). Right : Perplexity vs number of epochs of training LSTM on PTB (4 GPUs, total batch size = 20).

|  |  | Size | Gflop | SGD | RGC | qRGC |
|---|---|---|---|---|---|---|
| Cifar10 | ResNet44 | 2.65 | 0.20 | 7.48% | **7.17%** | 7.87% |
|  | VGG16 | 59 | 0.31 | 8.31% | 8.45% | **8.13%** |
| ImageNet | AlexNet | 233 | 0.72 | **44.73%** | 44.91% | 44.80% |
|  | ResNet50 | 103 | 8.22 | 24.07% | 23.98% | **23.85%** |
|  | VGG16 | 528 | 15.5 | 29.5% | **29.1%** | 29.3% |
| PTB | LSTM | 204 | 2.52 | 75.86 | **75.14** | 74.69 |
| Wiki2 | LSTM | 344 | 2.52 | 88.23 | 88.01 | **87.84** |

| Batch Size | 128 | 256 | 512 | 1024 | 2048 |
|---|---|---|---|---|---|
| ResNet44 |  |  |  |  |  |
| SGD | 7.09 | 7.48 | 8.18 | **10.02** | 16.8 |
| RGC | **6.40** | 7.17 | 7.471 | 10.13 | 10.87 |
| qRGC | 7.06 | 7.87 | 7.62 | 11.86 | **10.83** |
| VGG16 |  |  |  |  |  |
| SGD | 7.74 | 8.31 | **9.06** | **9.49** | 10.09 |
| RGC | **7.43** | 8.45 | 9.31 | 9.90 | 11.12 |
| qRGC | 8.17 | **8.13** | 9.09 | 9.97 | **9.81** |

Table 1: Results of RGC are achieved by non-quantized RGC method, and results of qRGC are achieved from quantized RGC method using RedSync. **The left table** : Accuracy results for various networks. Size indicates the model size in MB. GFlop shows Giga Floating-Point Operations required for a forward pass using a single input sample. Accuracy of CNNs was measured as top-1 validation errors, and accuracy of LSTMs is measured as perplexity on validating dataset. Results on Cifar10 were measured using 4 nodes with batch-size as 64 for each node. Results on ImageNet were measured using 6 nodes with batch-size as 32 for each node. Results of LSTM were measured using 4 nodes with batch-size as 5 for each node. **The right table**: Test errors of RCG and SGD methods under different batch sizes on Cifar10.

server with eight GPUs in the same node. It is equipped with one Intel(R) Xeon(R) CPU E5-2640 v4 and 8 TITAN Vs, which is connected to the CPU through PCI-E 3.0. **Piz Daint** is a GPU supercomputer. Each node of it includes two Intel Xeon E5-2690v3 CPUs and one NVIDIA Tesla P100 GPUs. In total, there are 5320 nodes connected by Aries interconnect with Dragonfly topology. We used pytorch v4.0 to conduct basic DNN training operations. For communication library, horovod an MPI wrapper upon pytorch, is used to provide collective communication operations. Horovod was compiled with OpenMPI v3.1 with cuda-aware supported on both systems.

We tested our performance on two major types of mainstream deep learning applications. For **Image Classification** tasks, we studied ResNet-44 and VGG16 on Cifar10 (Krizhevsky & Hinton (2009)), AlexNet, VGG16 and ResNet-50 on ImageNet (Deng et al. (2009)). For all CNNs, we used Nesterov's momentum SGD as optimizer. We used the same learning rate strategies as the SGD for the RGC methods. Warm-up technique was applied to the first 5 epochs of ResNet50 and VGG16 for both SGD and RGC. For **Language Modeling** tasks, we picked two datasets for evaluation. The Penn Treebank corpus (PTB) dataset consists of 923,000 training, 73,000 validation and 82,000 test words (Marcus et al. (1993)). The WikiText language modeling dataset is a collection of over 100 million tokens extracted from the set of verified Good and Featured articles on Wikipedia (Merity et al. (2016)). It consists 2,088,628 training, 217,646 and 245,569 test words. We adopted a 2-layer LSTM language model architecture with 1500 hidden units per layer (Press & Wolf (2016)) to evaluate both datasets. We tied the weights of encoder and decoder and use vanilla SGD with gradient clipping. Learning rate decays when no improvement has been made in validation loss.

## 3.2 EVALUATION OF ACCURACY

We examined the convergence of RedSycn on the datasets mentioned before. For the Cifar10 dataset, we used two CNNs, i.e. ResNet44 and VGG16, as test cases. Both DNNs were tested on 4 GPUs, and the total mini-batch size is 256. On the ImageNet dataset, we tested AlexNet, ResNet50, and

VGG16. On the PTB and Wiki2 dataset, we examined the perplexity of the 2-layer LSTM mentioned before.

Figure 2 shows the validation error of RGC and quantized RGC provided by RedSync on three test cases compared with original SGD. More comprehensive results are shown in the left side of Table 1. We also tested the sensitivity of the RGC method to large training data batch size. As shown in the right side of Table 1 when increasing the batch size to 2048, RedSync got no loss of accuracy compared to the original SGD.

## 3.3 Evaluation of Scalability and Speed

Next we tested the performance and scalability of RedSync as number of GPUs grow. Fig. 5 illustrates scalability of RedSync on Piz Daint with four test cases. Fig. 3 and Fig. 4 show the performance of RedSync on Muradin with six test cases. We compared *RedSync* and its quantization version *Quantized-RedSync* with a baseline data parallel implementation provided by horovod. Data was collected by averaging training time in 1000 training iterations. We used trimmed top-k algorithm to compress layers in CNNs larger than 128KB and used threshold binary search algorithm for hidden layers and the softmax layer for LSTM. Fig. 6 illustrates the cost of different parts using RedSync when scaling it to 128 GPUs on Piz Daint. Our observations are summarized as follows.

1. Using our parallel-friendly selection methods for compression is critical for system overall performance. In Fig. 3 and Fig. 4, we added an RGC implementation called *pure RGC*, which uses radixSelect to select top 0.1% elements as communication-set rather than our proposed methods. The performance of *pure RGC* is even slower than the baseline version, because compression time is too long.

2. RedSync is suitable for accelerating data parallel training on DNNs with high communication to computation ratio. For VGG16, AlexNet and LSTM, although performance of RedSync on a single GPU is not as good as baseline version due to compression and decompression overhead, RedSync can achieve significant speedup with more than 2 GPUs. However, we observed no performance gain for ResNet50 both on Piz Daint and Muradin. As implicated in Table 1, the ratio of computation to communication of ResNet50 is the highest in the DNNs we investigated. On large scale, most of time during ResNet50 training with RedSync is wasted on decompression phase, as shown in Fig. 6, which overdrafts the benefit of communication bandwidth reduction.

3. The scalability curve of RedSync on Piz Daint shows a concave shape. For example, as shown in Fig. 5, RedSync gets a better speedup to baseline version on 32 GPUs than 128 GPUs for AlexNet. It is because that communication bandwidth requirement and decompression overhead both grow linearly with the number of GPU in use. Such phenomenon verifies our analysis using communication performance model.

4. *Quantized-RedSync* always achieves better performance than *RedSync* for CNNs. However, for LSTM training on small scale, *Quantized-RedSync* achieves worse performance than *RedSync*. This is due to the balance of communication and computational overhead. CNN adopts trimmed top-k as the communication-set selection method and its quantized version has similar computation cost. As shown in Fig. 6, no significant difference of *selection* cost in CNN training. Therefore, the reducing of communication cost by quantization improves the system's overall performance. As for LSTMs, they use sampled threshold binary search as selection for non-quantized RedSync, but use threshold binary search for quantized RedSync. Sampled selection is much more faster. Therefore, on small-scale, *RedSync* has better performance than *Quantized-RedSync* due to less selection overhead. When scaling to more than 16 GPUs, benefit from the reduction of communication compensates for the cost of the communication-set selection.

## 4 Conclusion

This paper proposes a distributed implementation called RedSync to accelerate data parallel DNN training by utilizing a type of gradient sparsification method named as Residual Gradient Compression (RGC). We solved two major obstacles to implement RGC on multi-GPU systems : high overhead of compression using GPU and lack of support for collective communication implementation for sparse data structures. We tested the performance of RedSync on two GPU platforms,

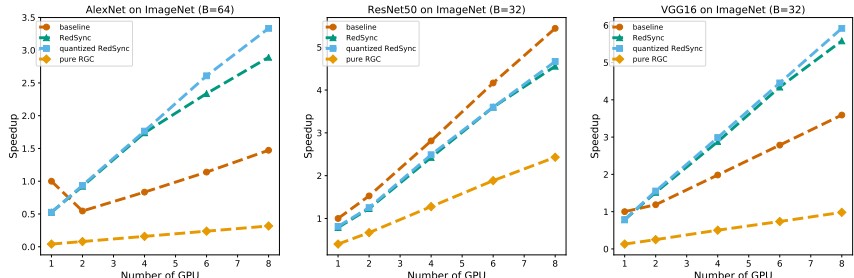

Figure 3: Scalability of RedSync for CNNs training on ImageNet using Muradin.

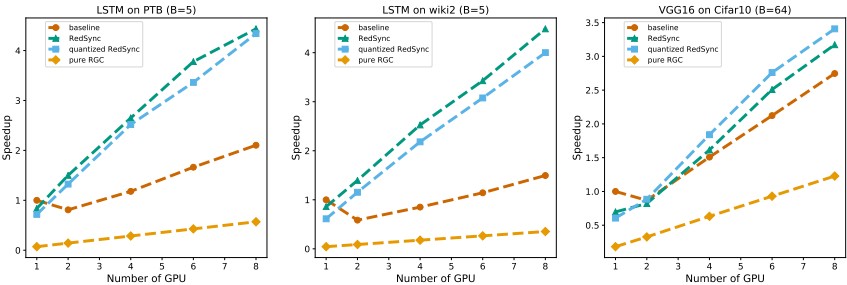

Figure 4: Scalability of RedSync for LSTM on PTB and Wiki2 datasets. Scalability of RedSync for LSTM VGG16 on Muradin.

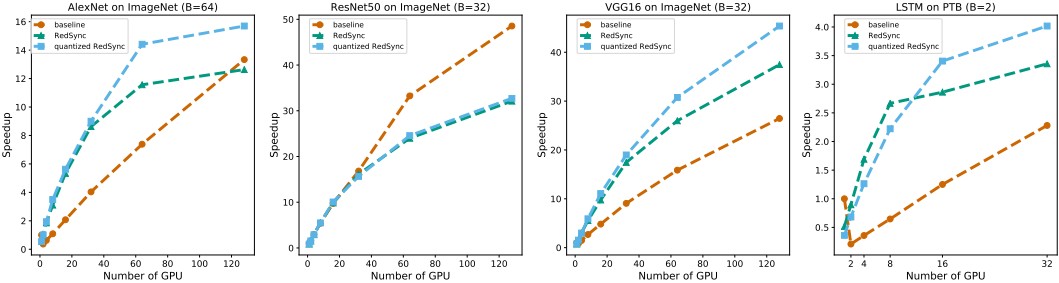

Figure 5: Scalability of RedSync for CNNs with ImageNet and LSTM with PTB on Piz Daint.

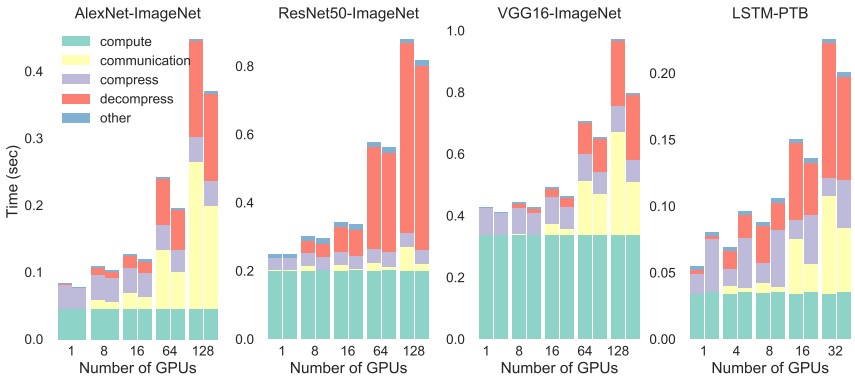

Figure 6: The time cost of different parts in RedSync on Piz Daint. Time is the average 10 iterations cost. For each two column group, the left column illustrates time decomposition for RedSync and right column illustrates time decomposition for quantized RedSync.

including a supercomputer system and a multi-GPU server. For AlexNet, VGG16, and LSTM, we observed significant speedup for large-scale DNN training.

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

## A   COST MODEL FOR SPARSE AND DENSE SYNCHRONIZATIONS

The left part of Figure 7 illustrates how sparse allgather works by recursive doubling method. We assume the compression rate on all of the node is the same as $D$. If we use *threshold binary search* for communication-set selection, $D$ here should be the average compression ratio of all nodes for a good approximation. In the first step, nodes that are a distance 1 apart exchange their compressed residuals, the size of which is $M \times D$. In the second step, nodes that are a distance 2 apart exchange their own data as well as the data they received in the previous step, which is $2M \times D$ in total. In the third step, nodes that are a distance 4 apart exchange their own data as well the data they received in the previous two steps. In this way, for a power-of-two number of processes, all processes get all the data in $\lg p$ steps. The amount of data exchanged by each node is $M \times D$ in the first step, $2M \times D$ in the second step, and so forth, up to $2^{lg(p)-1}M \times D$ in the last step. Therefore, The time for message transfer taken by this algorithm is $T_{transfer} = lg(p)\alpha + (p-1)M \times D\beta$. After including decompressing overhead $\gamma$ for collected $p$ different compressed residuals and communication selection overhead $T_{select}$, the time for all-gather based synchronization should be $T_{transfer} = T_{select} + lg(p)\alpha + (p-1)M \times D\beta + p\gamma_1$

As shown in the right part of Figure 7, the Rabenseifners algorithm is adopted for allreduce operation on messages. It does a reduce-scatter followed by an allgather. Reduce-scatter is a variant of reduce in which the result, instead of being stored at the root, is scattered among all $p$ nodes. We use a recursive halving algorithm, which is analogous to the recursive doubling algorithm used for allgather but in reverse way. In the first step, each node exchanges data with a node that is a distance $p/2$ away: Each process sends the data needed by all processes in the other half, which is of size $M/2$. They also receives the data needed by all processes in its own half, and performs the reduction operation on the received data. In the second step, each process exchanges data with a process that is a distance $p/4$ away. This procedure continues recursively, halving the data communicated at each step, for a total of $\lg p$ steps. After reduce-scatter, allgather phase will have the the same bandwidth and latency requirements. The time taken by Rabenseifners algorithm is the sum of the times taken by reduce-scatter (recursive halving), allgather and reduction operations. The total time should be $T_{transfer} = 2lg(p)\alpha + 2\frac{p-1}{p}M\beta + \frac{p-1}{p}M\gamma_2$.

## B   OVERLAPPING COMMUNICATION AND COMPUTATION

It is necessary to improve data parallel efficiency by overlapping communication with computation through pipelining communication and gradient calculation. Before updating aggregated gradients after scaling with learning rate to weights, gradient clipping is usually adopted to avoid gradient explosion. It rescales all of the gradients when the sum of their norms exceeds a threshold. For RGC methods, the local clipping technique (Lin et al. (2017)) is adopted to perform gradient clipping by a new threshold ($N^{-1/2}$ of original) locally before adding the current gradients to previous residuals. The difference is that traditional data parallel does clipping after communication of all layers are completed, while the RGC algorithm needs to do clipping before communication. In this case, we need to wait for the completion of the entire back-propagation to get gradients of all layers. And then we do clipping on gradients and then perform compression for communication. Local

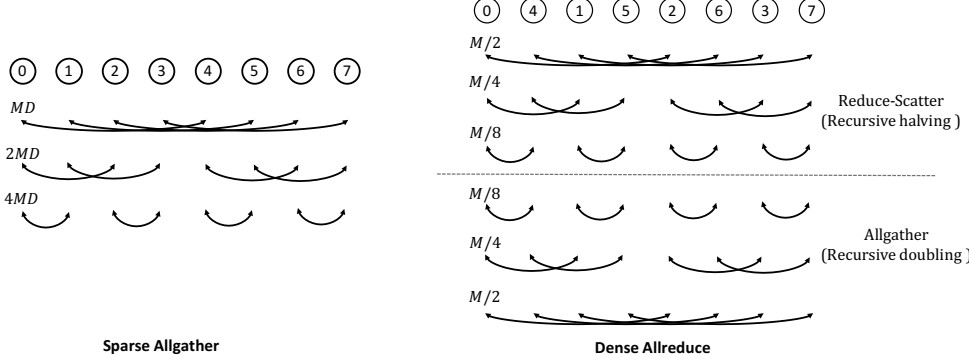

Figure 7: Communication pattern of sparse synchronization with allgather and dense synchronization with allreduce.

clipping is equivalent to introducing synchronization between computing and communication and thus eliminating the possibility of Communication hiding.

As shown in Figure 8, We have abandoned gradient clipping for CNNs, which seldom have gradient exploration problem for the deep networks in order to explore the potential overlapping. As for RNNs, gradients are achieved after backpropagation of all time steps using Back Propagation Through Time (BPTT). When backpropagation of the last layer is completed, we use the gradients of all layers to conduct local gradient clipping. In this case, the communication time can only overlap with the compression calculation. Because even with the original data parallel approach, the computation and communication overlap for each layer can only be made at the last time step, RGC dose not introduce too much overhead.

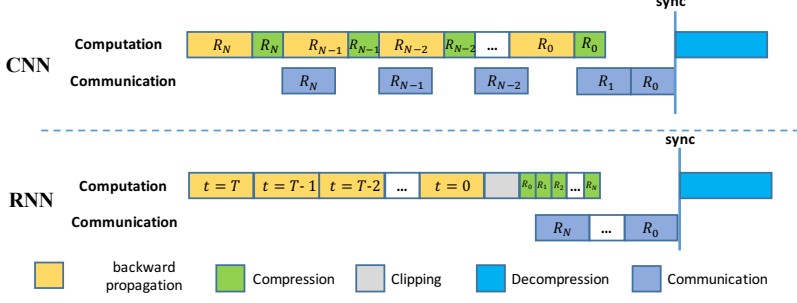

Figure 8: Two different schemes to overlap communication with computation for CNNs and RNNs.

## C  CORRECTNESS FOR MOMENTUM SGD AND WARM-UP TRAINING

We integrate the momentum masking and momentum correction schemes as proposed in Lin et al. (2017) for momentum SGD and Nesterov momentum SGD optimizers in RedSync. The Momentum SGD version of RGC method adopted by RedSync is illustrated in Algorithm 4. A warm-up training, by exponentially decreasing the compression ratio of the residuals in communication-set in first few epochs, is generally adopted to accelerate convergence in the first few iterations. For example, it is recommended to decrease the compression ratio of residuals in the warm-up period as follows: 25%, 6.25%, 1.5625%, 0.4%, 0.1%. However, we find it could be inefficient for large-scale. As analyzed in the previous section, even synchronization of compressed residual with a compression ratio as 1.5625% requires 100% bandwidth of dense allreduce for quantized RedSync on 64 GPUs. Instead of adopting high-compression-ratio RGC method of warm-up training, we use original SGD optimizer synchronized by allreduce in first few epochs if necessary.

---

**Algorithm 4** Residual Gradient Compression using MSGE

---

**Input:** node id $k$, the number of node $N$
**Input:** dataset $\chi$
**Input:** $use\_momentum$, $momentum$, $use\_nesterov$
**Input:** mini batch size $b$ per node
**Input:** initial model $w = w[0], ..., w[\#layer]$
**Input:** compression ratio $D$
   $V^k \leftarrow 0$
   $U^k \leftarrow 0$
  **for** $t = 0, 1, ...max\_iter$ **do**
     sample $b$ elements as $\chi_k^t$
     $G^k \leftarrow \nabla f(\chi_k^t ; \mathbf{w})$ by forward and backward propagation
     **for** $j = \#layer, \#layer - 1, ..., 0$ **do**
       **if** $use\_momentum$ **then**
         $U_j^k = momentum \cdot U_j^k + G_j^k$
         $V_j^k = V_j^k + U_j^k$
         **if** $use\_nesterov$ **then**
           $V_j^k = V_j^k + G_j^k$
         **end if**
       **else**
         $V_j^k = V_j^k + G_j^k$
       **end if**
       Masks $\leftarrow$ `selection` $(V_j^k, D)$
       $G_j^k \leftarrow$ `Allreduce(compress`$(V_j^k \cdot$ Masks$))$
       $V_j^k \leftarrow V_j^k \odot (1$ - Masks$)$
       **if** $use\_momentum$ **then**
         $U_j^k \leftarrow U_j^k \odot (1$ - Masks$)$
       **end if**
     **end for**
     $\mathbf{w} \leftarrow$ SGD$(\mathbf{w},$ `decompress`$(G^k))$
  **end for**

---

