# OpenReview forum: "RedSync : Reducing Synchronization Traffic for Distributed Deep Learning"
_ICLR.cc/2019/Conference_

### Official Review · AnonReviewer2 · 2018-11-02
**Good implementation optimizations in a important practical problem, but relatively incremental contribution**

**Rating:** 5
**Confidence:** 4

**Review:**

Quality and clarity:
The paper proposes an approach to reduce the communication bandwidth and overhead in distributed deep learning. The approach leverages on previous work (mainly the residual gradient compression (RGC) algorithm), and proposes several implementation optimizations. From what I can read, it is the basic RGC algorithm that is used, but with some clever optimization to improve the performance of it.

The quality of the paper is good, it is well-written and easy to read. The evaluation of the proposed approach is well done, using several diverse datasets and models, and executed on two different parallel systems. However, the reasons why RGC and qRGC sometimes have better accuracy than SGD needs to be analyzed and explained.

Originality and significance:
The originality of the paper is relatively low (optimization of an existing algorithm) and the contributions are incremental. However, the paper addresses an important practical problem in distributed learning, and thus can have a significant practical impact on how distributed deep learning systems are implemented.

Pros:
* Addresses an important issue.
* Good performance.
* Good evaluation on two different systems.

Cons:
* Limited contribution. Although I like implementation papers (very important), I think the contribution is to low for ICLR.

Minor:
* In general, the figures are hard to read (the main problem is to small text)
* Compression in the title is slightly misleading, since it's mainly selection that is done (top-0.1% gradients). Although the values are packed in a data structure for transmission, it's not compression in a information theory perspective.

---

> ### Author Response · Authors · 2018-11-20
> **Response**
>
> Thank you for your sincere comments.
> We add one sentence in the paper to clear that our contributions lie in system perspective rather than information theory perspective.
> We also have reorganized the figures and make them more clear.

---

### Official Review · AnonReviewer1 · 2018-11-03
**RedSync should implement a more systematic approach for optimization.**

**Rating:** 5
**Confidence:** 4

**Review:**

This paper introduces a set of implementation optimizations for minimizing communication overhead and thereby reducing the training time in distributed settings. The method relies on existing gradient compression and pruning techniques and is tested on synchronous/data-parallel settings.

The contribution and impact of the paper is unclear. The authors claim implementation innovations that show true performance gains of gradient compression techniques. But again it is unclear what those innovations are and how they can be reused for accelerating training for a new model.

The authors did perform an extensive set of experiments and while the method works well for some models and batch sizes, it doesn't work well for some other models. What would make the paper much more compelling would be if it came up with ways to systematically explore the relationship between training batch size, model parameter size, communication/computation/decompression ratos, and based on these properties, it can come up with best strategies to accelerate distributed data parallel training for any new model.

The paper needs to be polished as it has multiple typos.

---

> ### Author Response · Authors · 2018-11-20
> **Response**
>
> Thank you for your comments.
> Admittedly, dirty works we did overshadows our main contributions.
> I believe the value of this paper for ICLR is that it is one of few works considers gradient sparsification from the perspective of real system implementation. We would like to share with our peers some of our experiences, although looks not so remarkable.
> 1.The fast top-0.1 method on GPU.
> 2.Using allgather for sparse allreduce.
> 3.Details for parallel Local Gradient Clipping.
> Thank you for your advice. Considering the limitation of space, a systematic tuning will be left as our future work.
> My draft may not be very well-written due to limited time. We have polished it and fix most of the typos.

---

### Official Review · AnonReviewer3 · 2018-11-05
**Good analysis and provides empirical value of gradient compression**

**Rating:** 5
**Confidence:** 3

**Review:**

Paper focuses on Residual Gradient Compression (RGC) as a promising approach to reducing the synchronization cost of gradients in a distributed settings. Prior approaches focus on the theoretical value of good compression rates without looking into the overall cost of the changes. This paper introduces RedSync that builds on the existing approaches by picking the most appropriate ones that reduce the overall cost for gradient reduction without unduly focusing on the compression rate.
The paper does this by providing an analysis of the cost of RGC and also the limitations in scaling as the bandwidth required grows with the number of nodes. It also highlights the value of applying different algorithms in this process for compression and the benefits and issues with each.

Pros:
- Useful analysis that will help direct research in this area
- Shows that this approach works for models that have a high communication to computation ratio
- Provides a useful approach that works for a number of models

Cons:
- Positive experimental results are on models that are typically not used in practice e.g. AlexNet and VGG16
- Speedups shown on LSTMs don't see worthwhile to scale, and in practice a model-parallelism approach may scale better

Corrections:
- Typo in notes for Table 1 last sentence RCG => RGC
- Typo in first sentence in section 3.2: RedSycn => RedSync
- Section 3.3, #2 last sentence: maybe overdrafts => overshadows ?

---

> ### Author Response · Authors · 2018-11-20
> **Response**
>
> Thank you for your comments.
> RedSync performs better with the worse network.  Actually, platforms used to test our implement, a supercomputer, and a multi-GPU server, are equipped with the relative good inter-connected network. If testing on worse network fabric, like Ethernet and Wifi, ResNet will also gain a performance boost.
> LSTM is traditionally scaled with model parallelism. However, as we mentioned, data parallel is the easiest way to scale out with limited modifications of original serial code.  Part of work in (Lin et al 2018) also involves LSTMs.

---

### Public Comment · (anonymous) · 2018-10-26
**where is technical contribution**

Hi there,
     In this paper, you claim that you design a cost-efficient method for communication, but the core of the algorithms is already shown in Lin et al 2018 ICLR. So from the technical perspective, I didn't see anything new here.
     Then you combine the encoding technique which is well established in the past years, and nothing new in this paper.
     If you say the technical contribution is only for the implementation part, and then I think it is too weak for the contribution, or you can call it programming skills in real practice.
     Thanks for reading your paper.

---

> ### Author Response · Authors · 2018-10-28
> **Clarifications on technical contribution.**
>
> Thank you for reading our paper.
> The Gradient Compression idea was first proposed in 2014. Its ultimate goal is to accelerate the performance of data parallel training in real practice. A set of work including (Lin et al 2018) is devoted to solving the convergence problem of the algorithm. Based on their efforts, our work is devoted to solving the performance problem of the algorithm. Some of our innovations are critical to the successful application of this algorithm, which is a big concern for the industry. More importantly, we pointed out that some algorithmic improvements are not equal to system performance improvements.
> You may think the contribution of our work lies in the implementation part. However, "implementation issues, parallelization, software platforms, hardware” are indeed included in the relevant topics of this conference (ICLR 2019).
> Thanks again for your comments.

---

### Public Comment · (anonymous) · 2018-12-06
**Errors in algorithm 2 and 3?**

I have implemented algorithm 2 and 3.
In algorithm 2, I think line 6's condition should be nnz < k. Because increasing threshold will only make nnz even bigger.
Also in algorithm 3, line 10 and 12, I think ratio is the value that should be assigned to r and l respectively, or else this doesn't make any sense for a binary search.
Please confirm if my findings are correct or not. Thanks.

---

> ### Author Response · Authors · 2018-12-06
> **You are right!**
>
> Thanks for your reimplementation.
> Both of your findings are right. I will correct them.

---

### Meta-Review · Area_Chair1 · 2018-12-17
**limited contribution**

**Confidence:** 5
**Recommendation:** Reject

**Metareview:**

This paper proposed Residual Gradient Compression as a promising approach to reduce the synchronization cost of gradients in a distributed settings. It provides a useful approach that works for a number of models. The reviewers have a consensus that the quality is below acceptance standard due to practicality of experiments and lack of contribution.